# A Systematic Exposition of Methods used for Quantification of Heart Regeneration after Apex Resection in Zebrafish

**DOI:** 10.3390/cells9030548

**Published:** 2020-02-26

**Authors:** Helene Juul Belling, Wolfgang Hofmeister, Ditte Caroline Andersen

**Affiliations:** 1Laboratory of Molecular and Cellular Cardiology, Department of Clinical Biochemistry and Pharmacology, Odense University Hospital, 5000 Odense C, Denmark; hbelling@outlook.com (H.J.B.); hofmeister@health.sdu.dk (W.H.); 2Clinical Institute, University of Southern Denmark, Winsloewparken 25, 1. floor, 5000 Odense C, Denmark; 3Faculty of Health and Medical Sciences, DanStem, Novo Nordisk Foundation Center for Stem Cell Biology, 2200 København H, Denmark

**Keywords:** zebrafish, heart regeneration, cardiomyocyte proliferation, cardiac fibrosis, systematic review

## Abstract

Myocardial infarction (MI) is a worldwide condition that affects millions of people. This is mainly caused by the adult human heart lacking the ability to regenerate upon injury, whereas zebrafish have the capacity through cardiomyocyte proliferation to fully regenerate the heart following injury such as apex resection (AR). But a systematic overview of the methods used to evidence heart regrowth and regeneration in the zebrafish is lacking. Herein, we conducted a systematical search in Embase and Pubmed for studies on heart regeneration in the zebrafish following injury and identified 47 AR studies meeting the inclusion criteria. Overall, three different methods were used to assess heart regeneration in zebrafish AR hearts. 45 out of 47 studies performed qualitative (37) and quantitative (8) histology, whereas immunohistochemistry for various cell cycle markers combined with cardiomyocyte specific proteins was used in 34 out of 47 studies to determine cardiomyocyte proliferation qualitatively (6 studies) or quantitatively (28 studies). For both methods, analysis was based on selected heart sections and not the whole heart, which may bias interpretations. Likewise, interstudy comparison of reported cardiomyocyte proliferation indexes seems complicated by distinct study designs and reporting manners. Finally, six studies performed functional analysis to determine heart function, a hallmark of human heart injury after MI. In conclusion, our data implies that future studies should consider more quantitative methods eventually taking the 3D of the zebrafish heart into consideration when evidencing myocardial regrowth after AR. Furthermore, standardized guidelines for reporting cardiomyocyte proliferation and sham surgery details may be considered to enable inter study comparisons and robustly determine the effect of given genes on the process of heart regeneration.

## 1. Introduction

Heart failure (HF) is a global condition that has been estimated to affect more than 38 million people [1]. The pathology of HF includes a loss of cardiomyocytes (CM) that causes the heart to pump an insufficient amount of blood to the body. The most common cause of HF is myocardial infarction (MI) [2]. MI is defined as an area that has reduced or absent oxygen perfusion, which can cause the initiation of necrosis, apoptosis and autophagy leading to death of myocardial cells [2,3,4]. It is generally accepted that the mammalian heart cannot regenerate after ischemia or injury [5]. Thus, the compensatory mechanism after ischemia or injury, are assumed to favor fibrotic healing and hypertrophy instead of regeneration and cardiomyocyte proliferation [2,5]. Unfortunately, fibrotic healing may result in loss of heart contractility, leading to various complications and symptoms such as arrhythmias that also contribute to an increase in morbidity and mortality [5]. Today, only heart transplantation and palliative medication is offered as treatment of MI. Thus, a substantial effort is ongoing worldwide to develop regenerative therapies that can reduce mortality and improve life quality for millions of people [6].

## 2. The Regenerating Zebrafish Heart

In search for new treatment options for MI, researchers have focused their attention on certain lower vertebrates such as amphibians (e.g. axolotl, salamander) and some fish species such as zebrafish, that have shown the capacity to fully regenerate their hearts following myocardial injury. Even though, the zebrafish anatomy differs from the human heart by only having one ventricle and one atrium (Figure 1), several similarities exist. The zebrafish heart itself consists of a pericardial sac as also seen in humans, the bulbous arteriosus that can be compared to the human aortic arch, and the sinus venosus that is analogous to the human vena cava [7]. However, a dissimilarity, beside a lack of pulmonary vasculature, is that the zebrafish have an absence of a coronary artery network that is found in the human heart. Instead, zebrafish have small coronary capillary like vessels localized to the compact muscle layer, which has to be taken into consideration when performing heart injury models [8]. In 2002, Poss et al. [9] pioneered the field of zebrafish heart regeneration when demonstrating that the zebrafish heart fully regenerates following removal of ~10–20% of the ventricular apex (AR) (Figure 1A–C). Poss and colleagues showed that the zebrafish cardiac muscle had gained its full contractile function and normal appearance after 60 days post injury (dpi) [9,10]. Since then, other adult injury models such as cryoinjury, genetic cell ablation and hypoxia models have also been established. Whereas regeneration time and scar formation differs between injury types (i.e Apex resection (AR), cryoinjury, genetic ablation), the defining process of regeneration in various zebrafish injury models remains the same, and is the requirement of fully differentiated cardiomyocytes to revert to a developmental like state and re-enter the cell cycle in order to replenish the lost myocardium [11,12]. Overall, the AR model consists of different stages [6]. At 3–12 h dpi, a fibrin clot is formed that eventually dissolves, and localized apoptosis and necrosis occur, along with activation of the immune system. In parallel, the endocardial and epicardial cells become activated. At 3 dpi, myofibroblasts occupy the scar and secrete extracellular material, while endo- and epicardial cells undergo proliferation to cover the interior and exterior of the heart within the injury site, with epicardial cells actively invading the injury site later in the regeneration process [13]. The scar is transient, and while being dissolved, cardiomyocytes start proliferating, peaking at day 7–14 dpi and then migrating into the injury site. Hereafter, the myocardial wall will undergo thickening until the AR heart has become fully recovered at 60 dpi. In contrast, cryo-injury includes more substantial scar formation and requires a longer recovery time of 80-130 days [6]. Thus, given the ability of the zebrafish heart to regenerate relatively quickly following AR, the mechanisms underlying cardiomyocyte proliferation in AR hearts are of high interest for researchers to shed new lights on how to treat MI in humans.

Despite the rapid identification of proliferative signals since the introduction of the zebrafish model (For a detailed review refer to [6]), one current shortcoming in the field is that the techniques used for evaluation of heart regrowth and regeneration after AR are limited. This complicates a robust evaluation of genes and processes affecting zebrafish heart regeneration. To clarify this issue in detail and spur enthusiasm to develop new techniques for assessing zebrafish heart regeneration, we herein set out to perform a systematical literature analysis illuminating the quality and precision of the most commonly used methods for evidencing heart regrowth. We focused on studies exploiting the AR model since this injury type is the most prominent heart regeneration model studied in zebrafish.

## 3. Methods

### 3.1. Search Strategy

The systematical approach for literature search, study selection and data extraction follow the PRISMA guidelines. However, since our aim was to make a systematical assessment of existing literature regarding the methods used but not the reported results, we did not evaluate for risk of bias as otherwise recommended for meta-analysis etc. [14]. H.J.B. performed the systematical screening (Figure 2) in the Embase and Pubmed databases at 03/12/2018 and W.H. repeated the search at 04/11/2019. The search included three key MeSH and Emtree terms *“Heart regeneration”*, *“Cardiac regeneration”* and *“Zebrafish”*. The complete search strings in Pubmed (https://www.ncbi.nlm.nih.gov/pubmed/) and Embase (Ovid Interface; http://ovidsp.dc1.ovid.com.proxy1-bib.sdu.dk:2048/sp-3.33.0b/ovidweb.cgi) were as follows: *Pubmed Search* ((((((("journal article" [Publication Type]) AND zebrafish [MeSH Terms]) AND heart regeneration [MeSH Terms]) AND cardiac regeneration [MeSH Terms]) NOT "review" [Publication Type]) NOT "biography" [Publication Type]) NOT "editorial" [Publication Type]) NOT "comment" [Publication Type] and *Embase search* (Zebrafish and Heart regeneration and Cardiac regeneration).mp. not review.pt. Notably, MeSH and Emtree terms rely on annotation resulting in a two-three months delay from publication to annotation. Thus, we also checked literature using searches based on related free text terms, and hereby identified and included one article [15] that followed the criteria.

### 3.2. Study Selection

Search identified studies were imported into the systematic data management program Covidence [16] where duplicates were initially removed by the program and then screened manually. Then Covidence was used to screen first on title and abstract, and then in a second round by full-text screening against inclusion and exclusion criteria (Figure 2, Appendix A). Inclusion criteria accounted only for studies that had used adult zebrafish as a model for cardiac injury and evaluated cardiac regeneration. Only studies using injury types corresponding to ablation, cryoinjury or AR were extracted. Thus, studies using scratching, squeezing of the ventricle or hypoxia as heart injury were excluded from the search [17,18]. Methodical studies that primarily introduced a protocol for e.g. how to induce injury, were not included in the search [19,20]. Finally, we excluded studies if relevant information was lacking, if the full article was inaccessible online, or if a study was published in another language than English [21,22,23,24].

### 3.3. Extraction of Data

All data were extracted by H.J.B., and then re-checked independently by W.H. Any disagreements were solved by discussion between the disagreeing parties and by involving D.C.A. to solve the controversy. For each study, detailed data on injury type was first extracted and then in a second round of data extraction only AR studies were further evaluated for extraction on data concerning author research group and methods used for quantification of heart regeneration and regrowth. This included both methods related to reestablishment of the cardiomyocyte pool and the myocardium, as well as, measures of scarring and function. Moreover, we extracted count data and continuous data on days of injury, proliferation markers used, and measures as well as design relating to a cardiomyocyte proliferation index. By using the Pivot Table function in Microsoft excel, the identified data were categorized, sorted and counted. Data are either presented as count or continuous data.

## 4. Results

### 4.1. Included Studies on Zebrafish Heart Regeneration Following Apex Resection

A total of 34 studies from Embase and 155 from Pubmed were retrieved. Ten duplicates were identified, and the remaining 179 studies were screened against inclusion criteria based on title and abstract using Covidence (Figure 2). Further full text screening was then conducted for 82 articles, where approximately half (47 studies) used apex resection (AR), 27 exploited cryo-injury, and only 8 studies performed genetic ablation (Table 1).

The 47 studies that used AR as an injury type, were performed by 24 different research groups, with only three groups headed by Poss (11 studies), Belmonte (5 studies), and Kawakami (3 studies) publishing more than two studies (Table 2). We identified that the majority of AR studies (45/47) used qualitative analysis of heart sections for evaluating viable myocardium, where 8 of these extended this evaluation to include quantitative measures such as scarring measures (Table 2). 34 out of the 47 studies assessed more directly heart regeneration by analyzing cardiomyocyte proliferation, whereas 6 out of 47 studies performed functional heart analysis (Table 2).

### 4.2. Qualitative and Quantitative Analysis of Cardiac Outgrowth

A total number of 45 out of 47 articles used histology on sections for assessing heart regrowth and regeneration after AR (Table 2). The two articles [87,88] that did not use histology were focused on evaluating whether heart function as measured by echocardiography reflect heart regrowth. Thus, histology in general seems to be an accepted qualitative method for the evaluation of heart regeneration in zebrafish. Histology was performed either by Acid Fuchsin Orange G (AFOG; 23 studies), Masson’s Trichrome (MT; 5 studies), or Hematoxylin/Eosin (HE; 5 studies) or single reagents hereof [44]. Whereas HE [9,15,32,46,59] stains nucleic acids and proteins (nonspecifically e.g., cytoplasm and extracellular matrix) [97], AFOG and MT staining highlight the difference between muscle and collagen. The preference for AFOG staining may likely be explained by a higher sensitivity for Collagen in particular in zebrafish as suggested by Poss et al. [9]. Eight out of the 45 studies [9,29,36,66,73,89,90,94] evaluated the extent of heart regeneration by quantifying the amount of fibrosis on heart tissue sections in 2D where the area of collagen was compared to the total area of the ventricle.

### 4.3. Quantification of Cardiomyocyte Proliferation

34 studies out of the 47 studies used CM proliferation either alone or in combination with the above mentioned method as a measurement for quantifying heart regeneration [9,11,12,13,15,29,30,31,32,35,36,44,46,51,59,60,64,68,69,73,74,79,80,81,83,84,85,86,89,92,93,94,95,96]. Accordingly, expression of proliferating cell nuclear antigen (PCNA) (20 out of 34 studies [15,29,31,35,44,59,60,68,69,74,79,80,81,83,84,85,89,94,95,96]), phosphohistone-H3 (PHH3) (7 out of 34 studies [9,11,29,30,31,64,68]) as well as incorporation of 5-Bromo-2′-deoxyuridine (BrdU) (17 out of 34 studies [9,11,12,13,29,30,31,32,36,44,46,51,64,73,86,92,93]) or 5-Ethynyl-2′-deoxyuridine (EdU)(2 out of 34 studies [59,85]) have been used to assess CM proliferation (Table 3, Table 4 and Table 5) in combination with a cardiomyocyte marker (myocyte enhancer factor 2 (Mef2c), Myosin heavy chain 1 (MYH1), α-Sarcomeric Actin, or a transgene CM reporter (such as *cmlc2*::DsRed2). One study used Phalloidin to define cardiac cells and one study based their evaluation on structural characteristics. The Assessment of CM proliferation was mainly reported as a CM proliferation index in percentage or as an absolute number, whereas only a few studies rely on qualitative evaluations (Table 5). Yet, besides the use of different proliferation- and CM markers, studies were distinct in study designs with respect to BrdU/EdU exposure time and the number of injections (Table 5).

### 4.4. Analysis of Heart Function

The zebrafish heart, despite having only two chambers and a lack of pulmonary vasculature, shows a similar pattern of electrical activity and pump function as measured in humans when using electrocardiography (ECG) and echocardiography (ECHO), respectively. Yet, the parametric values are different between species, and data are thus mainly suited for relative measures between groups such as AR and sham zebrafish [88,98].

In our systematical analysis we identified a total number of six studies [7,12,49,87,88,89] that used a functional heart test as a measurement for cardiac recovery after AR. Three of these studies [7,49,89] used electrocardiography (ECG) while the other half [12,87,88] performed echocardiography.

The three studies that used ECG evaluated functional recovery after AR either by measuring variation in the R-R interval, (the time between each heartbeat) [89], or by measuring variation in the QT intervals [7,49].

## 5. Discussion

To the best of our knowledge, we here provide the first systematical analysis of methods used for evaluating heart regrowth and regeneration after apex resection in zebrafish. This is important since zebrafish offers an opportunity to identify mechanisms underlying heart regeneration, which may then be translated into the non-regenerative mammalian heart to improve heart repair after MI. 47 zebrafish studies performing AR were identified and showed that 95.7% of these studies used histology either alone or in combination with immunofluorescence detection of cardiomyocyte proliferation to assess zebrafish heart regeneration. 82.2% of the histology studies were merely qualitative, while 82.4% of the CM proliferation studies were quantitative. Only 12.8% of all studies included functional heart test, and no study used height or volume of the regenerating myocardium, which may provide even more robust evidence for heart outgrowth as discussed below.

Overall, histology was used in the studies to discriminate between viable myocardium and collagen rich scar tissue. The latter is a major outcome after MI in mammals, and as such it cannot compensate functionally for the CM loss. However, despite being informative on the presence or absence of fibrosis in the AR zebrafish hearts, such histology stains do not evidence heart outgrowth and regeneration capacity. This likely explains why only 8 out of 47 studies indeed perform quantitative histology (e.g. scarring index), whereas remaining studies only report qualitative evaluations. Moreover, one major caveat of this type of assay is that quantifications are based on a few sections, which then represent the whole heart. In this regard, it is important to note that consensus in the field exist, where the two-three largest sections of the ventricle are chosen to make an averaged and thus more robust analysis [87,95,96]. Yet, this could also bias heart outgrowth analysis, since one will select towards the size of the sections and not the injury site itself. With the numerous new imaging modalities 3D analysis or at least 2D serial analysis throughout the heart could be an alternative to minimize such bias and enable more robust quantifications as discussed below. Still the use of histology stains should be treated with cautions when claiming heart outgrowth. For instance, the degree of fibrosis and the amount of viable myocardium may depend on initial heart size and lesion performance, both parameters that may vary a lot in a heart that is only 1mm wide on average [6].

In contrast to fibrosis as a measure of heart regeneration, CM proliferation seems more appropriate as this biological process is considered the main underlying mechanism of heart regeneration [99]. As in general, zebrafish CMs go through the four phases of the cell cycle: G1, S, G2 and M, but in contrast to mammals the proportion of binucleated and polyploid CMs are <5% [36,99]. The latter may partly explain why zebrafish accomplish heart regeneration per se, but it is also important to consider when quantifying CM proliferation. In mammals, the majority of proliferation markers cannot distinguish whether a CM indeed undergoes division or simply binucleates or becomes polyploid (as reviewed elsewhere [100,101]), but in zebrafish it must be assumed that expression of these markers mainly represents the formation of two daughter CMs, although binucleation may occur at a low rate. However, the reported rates of CM proliferation may be difficult to compare across studies (Table 5), since the experimental design and reporting manner vary (Table 5). Also, the choice of cardiomyocyte- and proliferation markers as well as the day of analysis post injury are different between studies (Table 3, Table 4 and Table 5).

The inclusion of a cardiomyocyte marker is a prerequisite since also fibroblasts, hematopoietic cells as well as epicardial-, endocardial- and vascular cells proliferate following AR injury [6,8,102]. While Mef2c is a nuclear transcription factor, MYH1 and α-Sarcomeric Actin are cytoplasmic proteins present, and all can be used to label cardiomyocytes in the [103,104]. Visualization of these proteins mainly relies on antibody detection and thus depends on the quality of the reagents and protocols as in general. Since the used proliferation markers PCNA, BrdU/EdU, and PHH3 represent proteins residing in the nucleus, Mef2c co-localization may be more specific as compared to co-localizations with MYH1, α-Sarcomeric Actin, and the cardiac myosin light chain cmlc2 reporters that may require confocal imaging in 3D to validate single CM co-localization. The use of Phalloidin and CM structural characteristics [32,46] may be less robust in defining CMs. The used proliferation markers PCNA, BrdU/EdU, PHH3 are detected via immunofluorescence in the identified studies. Yet, it is important to note that they differ in their expression during the cell cycle which can have an impact upon analysis and interstudy comparison [100,101]. Pulse chase labelling using the thymidine analogues BrdU or EdU are golden standards for assessing cell proliferation through DNA labeling [105,106]. When comparing with PCNA, one must consider that PCNA marks cardiomyocytes from G_1_ phase until S phase, whereas thymidine analogues marks cardiomyocytes from S phase and onward. This will result in higher number of proliferating cardiomyocytes detected by PCNA per cell cycle [100,101,107]. In the identified studies, evaluation of CM proliferation was performed at different timepoints with 7- and 14 dpi being most commonly used (see Table 4 and Table 5) [9,60]. Since longer periods of BrdU/EdU incorporation and repetitive labelling pulses will increase the number of labelled cardiomyocytes, the quantifications depend on the specific BrdU/EdU setup [107] and should be taken into consideration upon interstudy comparisons.

In humans, the heart pump function is the ultimate determinant of a patient’s well-being after MI with arrythmias after infarction being a dangerous complication, and in mouse heart regeneration studies functional analysis such as MRI, ECHO or PET often are included. The similar pattern of electrical activity between zebrafish and humans despite the obvious anatomical difference makes it also a candidate for similar functional studies in zebrafish. However, in zebrafish AR studies, only 6 out of the 47 identified studies used functional analysis of heart function likely due to the small size of the zebrafish heart and the sensitivity of available techniques. ECHO is a non-invasive method in zebrafish, that uses ultrasound for imaging of cardiac structures and visualizes systolic and diastolic function [108], representing ventricular emptying and ventricular filling, respectively. In two [87,88] out of the three ECHO studies identified, ECHO was the sole measure of heart recovery and regeneration. In this regards it is important to acknowledge that ECHO is highly subjective and depends on user training and equipment quality among other parameters [109]. ECG, another non-invasive method, on the other hand might be more reproducible, and favorable for studying arrythmias, as indicated by QT- and T-wave abnormalities that can appear after AR. Prolongation of QT interval would suggest alterations in the ventricular repolarization that should normalize over time when the heart is healed and can thus be used as a readout for complete functional regeneration. Interestingly study showed a type of arrythmia with a prolongated QT interval after AR, reflecting ventricular repolarization and which is also seen after cryoinjury [49,51]. This may still persist even after complete heart regeneration as evidenced by histology [49]. In contrast, another article reported a shortened QT interval, another type of arrythmia after AR [7]. With only six studies using a functional readout and varying results, there is room for an improvement in implementation of techniques such as ECHO and ECG analysis in zebrafish.

## 6. Conclusions and Perspectives

Finally, we cannot emphasize enough that the present analysis is restricted to the data collected for the present study and we also cannot exclude that a few eligible studies have been unnoticed by our search strategy, and we apologize to authors of those studies. Other factors than those analyzed and discussed above may as well have an impact on the evaluation of zebrafish heart regrowth after AR. Particular we observed that only 3 out of 47 articles [46,49,74] explained the procedure of sham operations in detail and described clearly whether the sham was matched with a particular dpi specimen. Since, CM proliferation can be induced by injury or activation of tissues adjacent to the myocardium such as the epicardium [60,102], sham animals at a matched day seem obligate to consider when evaluating heart regeneration after AR.

Furthermore, our data suggest that some of the more advanced new imaging modalities could improve the field and accuracy of determining heart regrowth after AR. For instance, one may consider measuring the myocardial volume or height after AR and compare with a day matched sham. In this regard, the volume or height of myocardial tissue could ideally be quantified by two-photon imaging of a persistent cardiomyocyte reporter following tissue clearance. Very recently, Mercader and co-workers used a similar approach for analyzing Sox10+ CMs following cell specific ablation within the zebrafish heart [110].

Thus, although some consensus exists in the field, more standard guidelines for testing and reporting as well as the use of new less biased approaches could improve the field and provide more close evidence of heart regrowth after AR in zebrafish.

## Figures and Tables

**Figure 1 cells-09-00548-f001:**
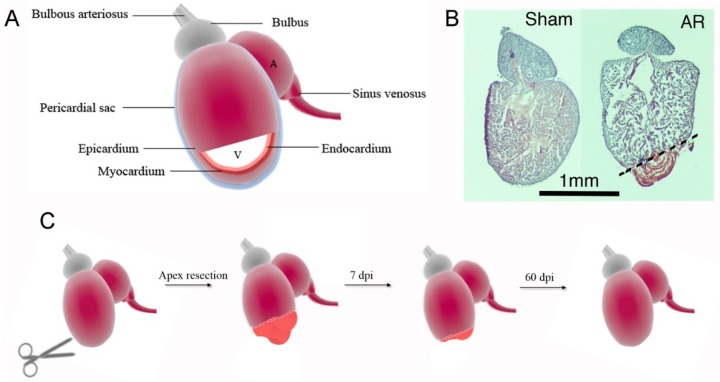
Apex resected (AR) zebrafish hearts regenerates during a 60 days time course. (**A**) The zebrafish heart consists of an atrium (A) and a ventricle (V). The cardiac layers, starting from the inner, is called endocardium, myocardium and epicardium. The heart is furthermore covered by a pericardial sac. The blood comes from the sinus venosus into the atrium and goes to the ventricle and out through bulbous arteriosus. (**B**) Hematoxylin staining of zebrafish hearts following sham surgery and AR of 10–20% of the heart. (**C**) Schematic representation of the overall processes occurring following AR in the zebrafish heart. Initially, a fibrin clot is formed in the apex region, whereafter cardiomyocytes starts to proliferate, peaking at 7–14 dpi, and regenerates the heart until 60 dpi, when it is fully recovered.

**Figure 2 cells-09-00548-f002:**
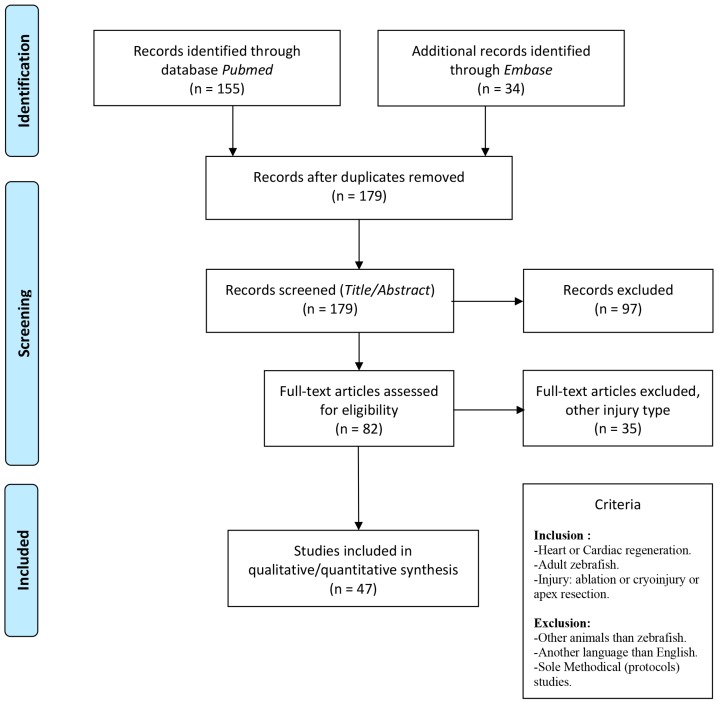
Flowchart of the literature search including inclusion and exclusion criteria. See methods and materials for details. As modified from PRISMA guidelines.

**Table 1 cells-09-00548-t001:** Systematically identified zebrafish studies divided by heart injury type.

Apex Resection (AR)	Cryoinjury	Genetic Ablation
Balciunas [25]	Azuaje [26,27]	Poss K.D. [28]
Belmonte [11,29,30,31,32]	Bakkers [33]	Mercarder [34]
Burns [35,36]	Butcher [37]	Poss [38,39,40,41,42,43]
Chuang [44]	Flores (CNIC) [45]	
Heideman [46]	Hassel [47,48]	
Hsiai [7,49]	Hsiai [50]	
Jazwinska [51,52]	Jazwinska [52,53,54,55,56,57,58]	
Kawakami [15,59,60]	Mercarder [61,62,63]	
Keating [64]	Planas [65]	
Kudo [66]	Pompa [67]	
Lee [68,69]	Stainer [70]	
	Reischauer S. [71]	
	Stainer [72]	
Lien [73]	Weidinger [74]	
Lou [75]	Cheng [76,77]	
Pedrazzini [78]		
Poss [9,12,13,38,79,80,81,82,83,84,85]		
Raya [86]		
Shih [87]		
Shung [88]		
Tsang [89,90]		
Wang [91]		
Weidinger [74]		
Xiong [92,93]		
Yin [94]		
Zhong [95,96]		

**Table 2 cells-09-00548-t002:** Methods used to evaluate heart regrowth and regeneration after apex resection (AR) in zebrafish.

Research Group (Last Author)	Qualitative Histology	Quantitative Histology	CM Proliferation	Heart Function
Total No.	45	8	34	6
Balciunas [25]	1			
Belmonte [11,29,30,31,32]	5	1	5	
Burns [35,36]	2	1	2	
Chuang [44]	1		1	
Heideman [46]	1		1	
Hsiai [7,49]	2			2
Jazwinska [51,52]	2		1	
Kawakami [15,59,60]	3		3	
Keating [64]	1		1	
Kudo [66]	1	1		
Lee [68,69]	2		2	
Lien [73]	1	1	1	
Lou [75]	1			
Pedrazzini [78]	1			
Poss [9,12,13,38,79,80,81,82,83,84,85]	11	1	9	1
Raya [86]	1		1	
Shih [87]				1
Shung [88]				1
Tsang [89,90]	2	2	1	1
Wang [91]	1			
Weidinger [74]	1		1	
Xiong [92,93]	2		2	
Yin [94]	1	1	1	
Zhong [95,96]	2		2	

**Table 3 cells-09-00548-t003:** Proliferation markers used to quantify cardiomyocyte proliferation after apex resection (AR) in zebrafish.

Research Group (Last Author)	PCNA	Brdu	PH3	Edu
Total No.	20	17	7	2
Belmonte [11,29,30,31,32]	2	5	4	
Burns [35,36]	1	1		
Chuang [44]	1	1		
Heideman [46]		1		
Jazwinska [51]		1		
Kawakami [15,59,60]	3			1
Keating [64]		1	1	
Lee [68,69]	2		1	
Lien [73]		1		
Poss [9,12,13,79,80,81,83,84,85]	6	3	1	1
Raya [86]		1		
Tsang [89]	1			
Weidinger [74]	1			
Xiong [92,93]		2		
Yin [94]	1			
Zhong [95,96]	2			

**Table 4 cells-09-00548-t004:** Timepoint (days past injury: dpi) used for assessment of cardiomyocyte proliferation after apex resection (AR) in zebrafish.

Research Group (Last Author)	3 dpi	7 dpi	10 dpi	14 dpi	30 dpi	60 dpi
Total No.	7	26	2	13	3	1
Belmonte [11,29,30,31,32]	1	3		3	2	
Burns [35,36]		1		2		
Chuang [44]			1			
Heideman [46]		1				
Jazwinska [51]		1				
Kawakami [15,59,60]	1	1		3		
Keating [64]	1	1		1		
Lee [68,69]		1		1		
Lien [73]		1	1			
Poss [9,12,13,79,80,81,83,84,85]	1	9		2	1	1
Raya [86]		1				
Tsang [89]		1				
Weidinger [74]	1	1				
Xiong [92,93]		2		1		
Yin [94]	1					
Zhong [95,96]	1	2				

**Table 5 cells-09-00548-t005:** Overview of design and reporting manners for cardiomyocyte proliferation studies after Apex resection (AR) in zebrafish.

Research Group	dpi	Exposure (Days)	No. of Injections	Proliferation- and Cardiomyocyte Markers	Proliferating CMs in AR
Belmonte [11]	14	7	7	No. BrdU+,GFP/Cmlc2+	400/section
Belmonte [31]	14	-	-	No. BrdU+/α-sarcomeric Actin+	3591
Belmonte [29]	14	14	7	No. BrdU+ cells (MyHC+)	250/section
Belmonte [32]	7	7+4 h *	7	Only visualization of BrdU+ (structural CMs)	-
Belmonte [30]	14	7+4 h *	7	No.BrdU+/α-sarcomeric actin+	750/section
Burns [35]	7	-	-	% PCNA+/Mef2+ cells out of total Mef2+	20%/in injury
Bruns [36]	14	7	1	% PCNA+/Mef2+ cells out of total Mef2+	7.5%/in injury
Chuang [44]	10	4	4	No. BrdU+ cells (Myosin+)	52/in injury
Heideman [46]	7	1	1	Only visualization of BrdU+/Phalloidin+	-
Jazwinska [51]	10	1 *	-	% BrdU+/Cmlc2:dsred+ out of Cmlc2:dsred+	4%/section
Kawakami [15]	14	-	-	No. PCNA+/Mef2+	21/section
Kawakami [59]	13	6	2	No. EdU+/Cmlc2-mCherry+	21/section
Kawakami [60]	7	-	-	No. PCNA/Mef2+	22/in injury
Keating [64]	14	7	1	BrdU+/Mef2+ (Only relative measures)	-
Lee [68]	7	-	-	% PCNA+/Mef2+ cells out of total Mef2+	15%/ in injury
Lee [69]	14	-	-	PCNA+/Mef2+/Troponin+	17%/unit area
Lien [73]	10	4 *	-	% BrdU+/Mef2+ cells out of total Mef2+	10%/ in injury
Poss [9]	7	7	1	%BrdU+/Myosin+ out of Myosin+	17%/ in injury
Poss [12]	7	3	3	Only visualization of BrdU+ cells	-
Poss [80]	7	-	-	% PCNA+/Mef2+ cells out of total Mef2+	12%/in injury
Poss [13]	7	3	3	Only visualization of BrdU+/cmlc2:nRFP+	-
Poss [79]	7	-	-	% PCNA+/Mef2+ cells out of total Mef2+	16%/ in injury
Poss [81]	7	-	-	% PCNA+/Mef2+ cells out of total Mef2+	17%/in injury
Poss [85]	7	-	-	% PCNA+/Mef2+ cells out of total Mef2+	15%/ in injury
Poss [83]	7	-	-	% PCNA+/Mef2+ cells out of total Mef2+	15%/in injury
Poss [84]	7	-	-	% PCNA+/Mef2+ cells out of total Mef2+	15%/unit area
Raya [86]	30	-	-	Only visualization of BrdU+/Myosin+	-
Tsang [89]	7	-	-	% PCNA+/Mef2+ cells out of total Mef2+	25%/in injury
Weidinger [74]	7	-	-	PCNA+ cells/cmlc2:GFP	12%/section
Xiong [93]	7	7	1	% BrdU+/Mef2+ cells out of total Mef2+	15%/in injury
Xiong [92]	14	7	1	% BrdU+/Mef2+ cells out of total Mef2+	15%/ in injury
Yin [94]	3	-	-	% PCNA+/Mef2+ cells out of total Mef2+	5%/ in injury
Zhong [95]	7	-	-	% PCNA+/Mef2+ cells out of total Mef2+	7%/ in injury
Zhong [96]	7	-	-	% PCNA+/Mef2+ cells out of total Mef2+	9%/ in injury

Only one selected timepoint and proliferation marker from each study is shown. The number or percentage of proliferating CMs represent a wildtype or a control (non-transgenic/non-manipulated). * Marks a design of BrdU added to the fish water instead of injection.

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
