# Peer review of "A Systematic Exposition of Methods used for Quantification of Heart Regeneration after Apex Resection in Zebrafish"

_cells, 2020, doi:10.3390/cells9030548_

Round 1

Reviewer 1 Report

The manuscript by Belling et al examines methods used in zebrafish heart regeneration experiments following apex resection.  The systematic and comprehensive analysis of the methods was performed to help the research community to identify best practices, evaluate quantitative methods and make recommendations for research going forward.  Overall, this is a useful analysis, which should stimulate discussion among the zebrafish heart regeneration community.

I have only minor concerns that are outlined below.

1. In most cases, the authors do not put a space between the number and units (e. g., 7dpi, which should read 7 dpi).

2. Line 90.  The word “paracrine” is used in reference to thyroid signals affecting heart regeneration.  Paracrine is short range signaling of a few cell diameters.  The thyroid signaling to the heart would be endocrine.  Please clarify.

3. Line 114.  Retinoic acid is written with and without a dash between the words.  The dash is not needed.

4. Line 357.  The authors discuss the use of 2-photon microscopy, but they add the word confocal.  Two-photon microscopy is not confocal, and the word should be deleted.

Author Response

REVIEWER 1

The manuscript by Belling et al examines methods used in zebrafish heart regeneration experiments following apex resection.  The systematic and comprehensive analysis of the methods was performed to help the research community to identify best practices, evaluate quantitative methods and make recommendations for research going forward.  Overall, this is a useful analysis, which should stimulate discussion among the zebrafish heart regeneration community.

I have only minor concerns that are outlined below.

Response: We are pleased that the reviewer finds our analysis useful for the field and agree to all suggested changes as described below. We thank the reviewer for his/her suggestions.

  1. In most cases, the authors do not put a space between the number and units (e. g., 7dpi, which should read 7 dpi).

Response: We apologize for this error and have corrected throughout including in tables.

  1. Line 90. The word “paracrine” is used in reference to thyroid signals affecting heart regeneration. Paracrine is short range signaling of a few cell diameters.  The thyroid signaling to the heart would be endocrine.  Please clarify.

Response: We agree, but since this paragraph has been removed as suggested by reviewer 2, the issue no longer exist.

  1. Line 114. Retinoic acid is written with and without a dash between the words. The dash is not needed.

Response: We agree, but since this paragraph has been removed as suggested by reviewer 2, the issue no longer exist.

  1. Line 357. The authors discuss the use of 2-photon microscopy, but they add the word confocal. Two-photon microscopy is not confocal, and the word should be deleted.

Response: We completely agree and have removed the word confocal.

Reviewer 2 Report

The authors aim to provide a systematic exposition of methods used for quantification of heart regeneration in zebrafish and in particular focus on the apical resection model. Even though I have great sympathy for this idea, which I think might provide a nice overview for people that are new to the field but also highlights discrepancies in previous studies and might even promote efforts to better harmonize future studies, I have several concerns. In particular, I think that the review remains superficial in many aspects but provides a great amount of (unstructured) detail in other parts. In general, it is very difficult to read and could benefit a lot from restructuring.   

Major:

Part 3 “Processes and biomolecules sustaining zebrafish heart regeneration” is a superficial description of several pathways involved in cardiac regeneration in the zebrafish. I see several problems with this part: i) It is very difficult to read, ii) it provides a great amount of information in a rather short paragraph, in particular readers that are not very familiar with heart regeneration in the zebrafish will not be able to follow, iii) on the other hand every single aspect is only mentioned briefly and the paragraph remains superficial and sometimes even resembles an enumeration ( e.g. Such as BMP signaling [14], Hedgehog signaling 88 [15,16] Nrg1/Erbb [17], Fgf [18] Jak1/stat3 pathway [19], Notch activation [20-22], TGF [15,23] NF-89 activation [24], Igf [25], Vitamin D [26] and retinoic acid signaling [27]. More recently paracrine thyroid signals have also been identified as inhibitory [28]). In addition, I am not sure how this paragraph contributes to the aim of the review.

The authors state in their discussion: “However, the reported rate of CM proliferation in the identified studies greatly varies (data not shown), which probably reflect the choice of cardiomyocyte- and proliferation markers as well as the day of analysis post 281 injury, all being different between studies (Table 3-4).” Why do the authors not show this information? This seems to be one of the most interesting aspects of their study.

The authors discuss in great length the use and limitations of proliferation markers. There are many reviews on this topic, e.g. from Felix Engel’s group and the review might benefit from a more focused view on the application of these markers in the zebrafish.

In contrast the authors provide only a very short and superficial discussion on the functional read outs, e.g. why does ecg, in particular qt-interval provide a meaningful measurement of regeneration.

Minor:

The authors state in the introduction that “MI is defined as an area that has reduced or absent oxygen perfusion…”. This is not the definition of a myocardial infarction. The author repeatedly mention the work from Kikuchi et al. and state that most cardiomyocytes that contribute to the regenerated heart muscle are gata4+. However, this study used a gata4+ reporter line but did not evaluate gata4 expression. There seems, at least to this reviewer, skepticism in the field that this reporter line does truthfully mirror gata4 expression.

Author Response

REVIEWER 2

The authors aim to provide a systematic exposition of methods used for quantification of heart regeneration in zebrafish and in particular focus on the apical resection model. Even though I have great sympathy for this idea, which I think might provide a nice overview for people that are new to the field but also highlights discrepancies in previous studies and might even promote efforts to better harmonize future studies, I have several concerns. In particular, I think that the review remains superficial in many aspects but provides a great amount of (unstructured) detail in other parts. In general, it is very difficult to read and could benefit a lot from restructuring.  

Response: We thank the reviewer for his/her constructive criticism of our work and agree to the majority of raised issues. We have thus made the suggested corrections and hope that the reviewer finds the manuscript sufficiently improved.

Major:

Part 3 “Processes and biomolecules sustaining zebrafish heart regeneration” is a superficial description of several pathways involved in cardiac regeneration in the zebrafish. I see several problems with this part: i) It is very difficult to read, ii) it provides a great amount of information in a rather short paragraph, in particular readers that are not very familiar with heart regeneration in the zebrafish will not be able to follow, iii) on the other hand every single aspect is only mentioned briefly and the paragraph remains superficial and sometimes even resembles an enumeration ( e.g. Such as BMP signaling [14], Hedgehog signaling 88 [15,16] Nrg1/Erbb [17], Fgf [18] Jak1/stat3 pathway [19], Notch activation [20-22], TGF [15,23] NF-89 activation [24], Igf [25], Vitamin D [26] and retinoic acid signaling [27]. More recently paracrine thyroid signals have also been identified as inhibitory [28]). In addition, I am not sure how this paragraph contributes to the aim of the review.

Response: This paragraph was indeed only meant as a quick introduction to the field, but we acknowledge the reviewer’s point and agree that the paragraph somehow diverge from the actual aim of the study. We have therefore deleted this part and thank the reviewer for pointing out this issue to us.

The authors state in their discussion: “However, the reported rate of CM proliferation in the identified studies greatly varies (data not shown), which probably reflect the choice of cardiomyocyte- and proliferation markers as well as the day of analysis post 281 injury, all being different between studies (Table 3-4).” Why do the authors not show this information? This seems to be one of the most interesting aspects of their study.

Response: We agree with the reviewer that this is an important point. However, as also stated in the methods. Since our aim is to make a systematical assessment of existing literature regarding the methods used but not the reported results, we did not evaluate for risk of bias as otherwise recommended for meta-analysis etc. Thus, we do not believe that we are entitled to include a complete overview of reported CM proliferation data and further discuss these. Instead we have generated a new Table 5, which gives an overview of the different ways of reporting and analyzing the CM proliferation based on one example from each study including the reported percentage or absolute number of proliferating CMs in wildtype or a control (vehicle control/non-manipulated). Although the readers are free to use this proliferation index for evaluating CM proliferation following AR in general, we do not comment as such on the reported values as this would be inappropriate and not the intention of the study.

We hope the reviewer agrees with us.

The authors discuss in great length the use and limitations of proliferation markers. There are many reviews on this topic, e.g. from Felix Engel’s group and the review might benefit from a more focused view on the application of these markers in the zebrafish. In contrast the authors provide only a very short and superficial discussion on the functional read outs, e.g. why does ecg, in particular qt-interval provide a meaningful measurement of regeneration.

Response: We have carefully re-read this part and have tried to shorten it while referring to the work by the Engel group. However, we believe that some knowledge on the markers are important to emphasize the difficulties when comparing CM proliferation between studies of different designs, especially for newcomers in the field. Moreover, we have inserted more discussion on the functional read outs and why it is important.

Minor:

The authors state in the introduction that “MI is defined as an area that has reduced or absent oxygen perfusion…”. This is not the definition of a myocardial infarction. The author repeatedly mention the work from Kikuchi et al. and state that most cardiomyocytes that contribute to the regenerated heart muscle are gata4+. However, this study used a gata4+ reporter line but did not evaluate gata4 expression. There seems, at least to this reviewer, skepticism in the field that this reporter line does truthfully mirror gata4 expression.

Response: We acknowledge the reviewer’s point, but since this paragraph has been removed as suggested, the issue no longer exists.

Round 2

Reviewer 2 Report

The article is a lot easier to read and the new table 5 is interesting. I only have two minor comments on the introduction.

"MI is defined as an area that has reduced or absent oxygen perfusion"

Today, only heart transplantation and palliative medication is offered as treatment of MI.

These statements are wrong and should be corrected. For MI definition please see “Fourth universal definition of myocardial infarction” (2018). What do the authors mean by “oxgen perfusion”? Perfusion with oxygenated blood? Interventional therapies are highly effective for acute MI and pharmacotherapy has substantially improved the prognosis for patients with coronary artery disease and heart failure. These are not major points for the manuscript but wrong descriptions of basic pathophysiology and (pharmaco)therapy in the introduction does not make a good impression for an otherwise nice manuscript.

Author Response

We agree with the reviewers point and have rephrased both sentences. We hope the wording is more accurate now and thank the reviewer for pointing this to us.